# Spinal pain patients seeking care in primary care and referred to physiotherapy: A cross-sectional study on patients characteristics, referral information and physiotherapy care offered by general practitioners and physiotherapists in France

Anthony Demont[1,2]*, Leila Benaïssa[3], Valentine Recoque[3], François Desmeules[4,5], Aurélie Bourmaud[1,2]

1 Université Paris Cité, Inserm, ECEVE, Paris, France, 2 AP-HP, Hôpital Robert Debré, Paris, France, 3 Physiotherapy School, University of Orléans, Orléans, France, 4 School of Rehabilitation, Faculty of Medicine, University of Montréal, Montréal, Québec, Canada, 5 Maisonneuve-Rosemont Hospital Research Center, University of Montreal Affiliated Research Center, Montréal, Québec, Canada

* anthony.demont@gmail.com

## Abstract

### Objectives

To describe spinal pain patients referred by their treating general practitioners to physiotherapy care, examine to which extent physiotherapy interventions proposed by general practitioners and physiotherapists were compliant to evidence based recommendations, and evaluate concordance between providers in terms of diagnosis and contraindications to physiotherapy interventions.

### Methods

This study included spinal pain patients recruited from a random sample of sixty French physiotherapists. Physiotherapists were asked to supply patients' physiotherapy records and characteristics from the general practitioner's physiotherapy referral for the five new consecutive patients referred to physiotherapy. General practitioner's physiotherapy referral and physiotherapists' clinical findings characteristics were analyzed and compared to evidence-based recommendations using Chi-squared tests. Cohen's kappas were calculated for diagnosis and contraindications to physiotherapy interventions.

### Results

Three hundred patients with spinal pain were included from sixty physiotherapists across France. The mean age of the patients was 48.0 ± 7.2 years and 53% were female. The most common spinal pain was low back pain (n = 147). Diagnoses or reason of referral formulated by general practitioners were present for 27% of all patients (n = 82). Compared to general practitioners, physiotherapists recommended significantly more frequently recommended

**Data Availability Statement:** All relevant data are within the paper and its Supporting information files.

**Funding:** The funders had no role in study design, data collection and analysis, decision to publish, or preparation of the manuscript.

**Competing interests:** The authors have declared that no competing interests exist.

**Abbreviations:** CI, confidence interval; CPG, clinical practice guideline; GP, general practitioner; MSD, musculoskeletal disorder; PT, physiotherapist; SD, standard deviation.

interventions such as education, spinal exercises or manual therapy. General practitioners prescribed significantly more frequently passive physiotherapy approaches such as massage therapy and electrotherapy. The overall proportion of agreement beyond chance for identification of a diagnosis or reason of referral was 41% with a weak concordance (κ = 0.19; 95%CI: 0.08–0.31). The overall proportion of compliant physiotherapists was significantly higher than for general practitioners (76.7% vs 47.0%; p<0.001).

## Conclusions

We found that information required for the referral of spinal pain patients to physiotherapy is often incomplete. The majority of general practitioners did not conform to evidence-based recommendations in terms of prescribed specific physiotherapy care; in contrast to a majority of physiotherapists.

## Trial registration

**ClinicalTrials.gov**: NCT04177121

## Introduction

Musculoskeletal disorders (MSKDs) are a major public health concern worldwide and represent globally the second most important group of disorders in terms of years lived with disability [1]. MSKDs account for approximately 17.0% of French general practitioners (GPs) consultations [2, 3] and their incidence is expected to increase as the French population is ageing rapidly [4]. The most common MSKDs encountered in primary care are spinal pain including neck, thoracic spine or low back disorders [5, 6]. In several health care systems, GPs are first-contact providers for patients seeking care for spinal pain complaints. GPs have a key role in the patient's care pathway by providing an initial diagnosis and treatment and referring to other appropriate health care professionals such as physiotherapists (PTs) [7]. The purpose of this referral is to validate the indication for physiotherapy care and to identify any contraindications or precautions to rehabilitation for a specific patient [7]. However, several studies have concluded that diagnoses provided by GPs for this population may often be erroneous or not as accurate as those provided by other MSKD specialists such as sports physicians, orthopedic surgeons or even PTs [8–10]. In addition, GPs' practice patterns in terms of treatment recommendations have been reported to divert significantly from established evidence-based clinical practice guidelines (CPGs); poor patient education as well as poor promotion of active treatments such as physical activity is often reported [3, 11–17]. Although concerns have been raised for initial care provided by GPs in primary care, PTs rehabilitation care for spinal pain patients is often cited also as not complying to evidence-based recommendations [18]. It is important that adequate care for these populations is efficient and patient-centered to limit deleterious consequences such as delay in treatment and potential clinical complications such as pain chronicization [11, 14, 19].

French GPs' physiotherapy referral practices for patients with spinal pain complaints in primary care have not been described and reported so far. Thus, the extent to which French GPs and PTs practices as regards to physiotherapy interventions prescribed, are supported by evidence-based recommendations of CPGs is not known. This study aimed, based on a sample of patients referred by GPs for physiotherapy to French licensed PTs working in private practice:

1- to describe types of spinal pain patients referred by their treating GPs to participating PTs, based on information collected from the GPs' physiotherapy referral form; 2- based on information on the GPs' physiotherapy referral, to examine to which extent, when specific physiotherapy interventions are prescribed by GPs, they adhere to evidence-based recommendations for care of these patients; 3- based on information in the patient's physiotherapy record, to examine to which extent physiotherapy interventions provided by the treating PT adhere to evidence-based recommendations for care of these spinal pain patients, and 4- to compare and evaluate concordance between information provided by the GP from the physiotherapy referral and the treating PT after their initial consultation on diagnosis and prescribed physiotherapy interventions.

## Materials and methods

### Study design

This study is a cross-sectional multicentered observational study including a sample of patients with spinal pain complaints initially referred by their treating GPs and recruited from a random sample of 60 French licensed PTs. This study conforms to all reporting items of the Strengthening the Reporting of Observational studies in Epidemiology checklist (STROBE) (see S1 Table in S1 File) and of the Guidelines for reporting reliability and agreement studies (GRRAS) (see S2 Table in S1 File). Ethics approval was obtained from the Ethics Committee of the Robert Debré Hospital (2019/441-2).

### Setting

French licensed PTs working in private practice were identified and selected, using a computer-generated random number list, from the list of all registered members of the French National Council of Physiotherapists. Based on the French law, patients with spinal pain seeking physiotherapy care cannot access a PT directly. They require a prescription from a physician to refer to a PT whose care will be covered by the French National Health Insurance. The inclusion of participating PTs was stratified by the 13 geographical regions in order to represent all French geographical regions. According to the geographical density of French PTs, four to five PTs per region were therefore identified and recruited. Recruitment took place between November 2019 and July 2020.

### Eligibility criteria for participating physiotherapists and for the sample of spinal pain patients

PTs inclusion criteria were: 1) licensed and working in a private practice in France, 2) to be registered with the French National Health Insurance, and 3) receiving and treating adult patients with spinal pain complaints referred from GPs. The only exclusion criteria was treating pediatric patients, aged 17 years old or younger.

The sample of referred spinal pain patients was formed from the five most recent patients with spinal pain and newly referred by their treating GP to each participating PT. Inclusion criteria for these participants were: 1) being an adult patient initiating a new episode of care with the PT and 2) newly referred by their GP for a spinal pain complaint of the neck, thoracic spine or low back regions. All participating PTs provided written informed consent at enrollment. Participating PTs were asked to supply the patients' physiotherapy record and the treating GP's physiotherapy referral for the five consecutive patients newly referred to physiotherapy. All patients also signed a consent form to allow access to their physiotherapy record.

## Participating French licensed physiotherapists demographic and professional characteristics

A standardized form, developed by a multidisciplinary team (two PTs, one GP, one trial methodologist, one statistician, and one sociologist) and pre-tested with five voluntary PTs, was provided to the participating PTs. This form was used to collect demographic and professional characteristics of the included PTs such as gender (male or female), age (in years), clinical practice location (rural or metropolitan), professional experience (in years), postgraduate training in spinal pain rehabilitation (yes or no) and graduating year for initial PT diploma. This last variable was categorized according to the three main reforms related to the French initial training curricula in physiotherapy (1946, 1989, and 2015).

## Spinal pain patient sample

Data extraction from the patient's physiotherapy record of the five new consecutive patients included by each participating PT was done by two authors (AD and LB). Demographic characteristics such as gender (male or female), age (in years), wait time between GP's referral and initial PT consultation (in days), the spinal pain area (neck, thoracic spine, low back or in combination—defined as concomitant neck pain, thoracic spine pain, and/or low back pain), presence of pain lasting more than three months (yes or no), the worst spinal pain intensity reported by the patient as measured during the initial PT consultation with a Visual Analog Scale (0–10), presence of lower limb referred pain (yes or no), the number of comorbidities per participant such as osteoporosis, cardiovascular or cerebrovascular disease were extracted from the patient's medical record, and if the reason for consultation was a work-related injury (yes or no). Discrepancies between the two evaluators (AD and LB) were resolved via discussion; a third evaluator was involved if no consensus was reached (AB).

## General practitioner's physiotherapy referral characteristics

For each patient included, the following physiotherapy referral characteristics prescribed by the GP were extracted by two independent authors (AD and LB) from the standardized prescription form used in clinical practice, when these characteristics and information have to be reported: 1) description of the involved anatomical region (neck pain, thoracic pain, low back pain or combination); 2) presence of a specific diagnosis or reason of referral; 3) indication of contraindications to certain physiotherapy interventions; 4) presence of related medical information provided with the referral such as imaging or other diagnostic test results; 5) information on types of physiotherapy interventions prescribed; 6) specific number of prescribed physiotherapy consultations; and 7) specific prescribed frequency per week of physiotherapy consultations. The types of physiotherapy interventions prescribed by GPs were categorized according to categories presented in selected CPGs and systematic reviews for the management of spinal pain adults [20–24] and included: 1) postural and hygienic education (such as advice on daily physical activity); 2) specific spinal exercises; 3) McKenzie exercises; 4) stretching exercises; 5) general exercises; 6) manual therapy; 7) massage therapy; 8) hot/cold therapy; 9) electrotherapy; and 10) ultrasound therapy. Discrepancies between the two evaluators (AD and LB) which performed the categorization were resolved via discussion; a third evaluator was involved if consensus was not achieved (AB).

## Physiotherapy initial consultation and treatment recommendations

Each spinal pain patient referred by the GP to physiotherapy was assessed by his/her participating PT as part to their usual clinical care. After the initial consultation with the patient, the

PT completed a standardized form to indicate: 1) anatomical region for spinal pain complaint of the patient; 2) specific working diagnosis; 3) contraindications to certain physiotherapy interventions; 4) types of physiotherapy interventions recommended; 5) specific number of recommended physiotherapy consultations; and 7) specific recommended frequency per week of recommended physiotherapy consultations. For the physiotherapy treatment recommended by PTs, their options were extracted and categorized using the same classification used for GPs.

## Determination of recommended care based on evidence-based clinical practice guidelines

To establish whether prescribed physiotherapy interventions either by the GP, based on the physiotherapy referral or the PT after their initial consultation, were consistent with evidence-based recommendations, physiotherapy interventions were compared to recommendations from selected evidence-based recommendations of CPGs for the management of neck pain, thoracic spine pain, and low back pain. Relevant recommendations from French CPGs published, and if not available, recent evidence-based recommendations of CPGs or systematic reviews published for the management of adults with spinal pain were extracted by two independent authors (AD and LB). Recommendations of one French CPG on the management of low back pain [20, 21], and two systematic overviews of current evidence for the management of neck pain [22, 23], and two on thoracic spine pain [23, 24] were selected. A summary of the recommendations extracted from the selected CPGs and reviews is shown in the S3 Table in S1 File [20–24]. In the absence of recent French CPGs for the management of patients with neck or thoracic spinal pain, we chose international guidelines [22–24], based on the highest level of scientific evidence from various competent authorities recognized internationally for the quality of their scientific productions. These CPGs are not specific to physiotherapists but to all primary care health professionals taking care of these populations of patients. In France, GPs must keep informed of the latest published medical evidence in order to adapt their practices, through professional development. This is a requirement of best medical practice for all medical doctors. The different specific physiotherapy interventions prescribed by GPs and recommended by PTs were categorized following a standardized manner by two independent authors (AD and LB) according to the most appropriate category of physiotherapy interventions from the selected CPGs; a third evaluator was involved if consensus on the most appropriate physiotherapy intervention category was not reached (AB) (S3 Table in S1 File). For each spinal pain participant, if at least one physiotherapy intervention prescribed by the GP or the PT was based on a recommendation of strong or moderate level from any of the selected CPGs or reviews, the provider (GP or PT) was then categorized as a compliant provider offering recommended care for that particular patient.

## Analyses

Categorical variables were expressed as frequency, percentages and number of missing data, and continuous variables as means, standard deviations and number of missing data. Categorical data on care prescribed by GPs or recommended by PTs after their initial consultation were analyzed and compared using the Chi-squared tests or Fisher's exact tests. Mann-Whitney-Wilcoxon tests were used to compare number of prescribed or recommended physiotherapy consultations for GPs and PTs as well as to compare between providers frequency per week of physiotherapy consultations. Two-sided alpha level was set at 0.05. Cohen's Kappa ($\kappa$) with associated 95% confidence intervals (95% CI) were calculated between each specific diagnosis and contraindications to certain physiotherapy interventions collected from GPs' physiotherapy referral and from PT's findings at the initial consultation. The working diagnosis of

the PT was considered as the reference standard; evidence report that diagnoses formulated by PTs are more accurate than those formulated by GPs for patients consulting with spinal pain [8–10]. Concordance values, reported as an estimate of agreement beyond chance, were interpreted according to the following criteria: κ = 0–0.20 weak, κ = 0.21–0.40 slight, κ = 0.41–0.60 agreement, κ = 0.61–0.80 high, κ = 0.81–0.90 very high or κ > 0.90 excellent agreement [25]. Due to the multiple possible diagnoses as well as the different nomenclature sometimes used by the GPs or PTs, diagnoses were put into diagnostic categories to establish if diagnoses were concordant. Generic diagnostic coding was performed by two independent authors (AD and VR) both for GPs and PTs, from the diagnoses reported by each of these providers when present. This was to avoid ontological differences as well as medical versus working specificities. The objective was to ensure that diagnoses provided by GPs and PTs were comparable. The following categories were used based on CPGs and systematic overviews selected: non-specific neck pain, cervical radiculopathy, specific neck pain, non-specific thoracic spine pain, non-specific low back pain, radiculopathy/sciatica, specific low back pain, or combination of concomitant spinal pain with concomitant diagnosis [21–24] and were determined by two independent reviewers (AD and VR). To measure the overall raw agreement beyond chance between providers for specific physiotherapy interventions prescribed or recommended by providers, authors defined that the presence of at least two concordant physiotherapy interventions was considered perfect agreement. The combination of physiotherapy interventions is frequently recommended by selected CPGs and systematic reviews for the management of spinal pain [21–24]. An independent third rater (AB) was consulted if consensus could not be reached. All statistical analyses were performed using the Statistical Package for the Social Sciences (SPSS) version 25 (IBM Corp Armonk).

## Results

### Participating French licensed physiotherapists and spinal pain patients' characteristics

During the seven-month data collection period (01/11/2019 to 31/05/2020), 138 invitations to participate were sent to eligible PTs, in order to get the final sample consisting in 60 participating PTs (Fig 1). Characteristics of the participating PTs are presented in Table 1. Participating PTs had a mean age of 38.1 years (SD: ±10.5) with 62% of men and 38% of women. The majority of PTs did not report any postgraduate training in spinal pain rehabilitation (58%; n = 35) and the majority graduated between 1990 and 2015 (65%; n = 39).

From the caseload of participating PTs, a total of 300 eligible patients gave access to their physiotherapy record and were included in the study; no patient refused participation and access to their record. Characteristics describing the participants are presented in Table 2. Patients had a mean age of 48.0 years (SD: ± 7.2), 47% were men and 53% were women. Mean waiting time between GP's referral and initial PT consultation was 12.4 days (SD: ± 6.2). The spinal pain involved region was the neck (16%; n = 47), thoracic spine (5%; n = 16), low back (49%; n = 147) or involved more than one area (30%; n = 90). Forty-seven percent of all patients reported pain lasting more than 3 months (n = 142). Mean worst pain intensity reported by the patient during the initial PT consultation and assessed with a Visual Analog Scale (0–10) was 7.0/10 (SD: ± 2.2).

### General practitioners' physiotherapy referral characteristics

Description of type of information provided by GPs from the physiotherapy referral of all spinal pain patients (n = 300) is presented in Table 3. 164 individual GPs were identified as

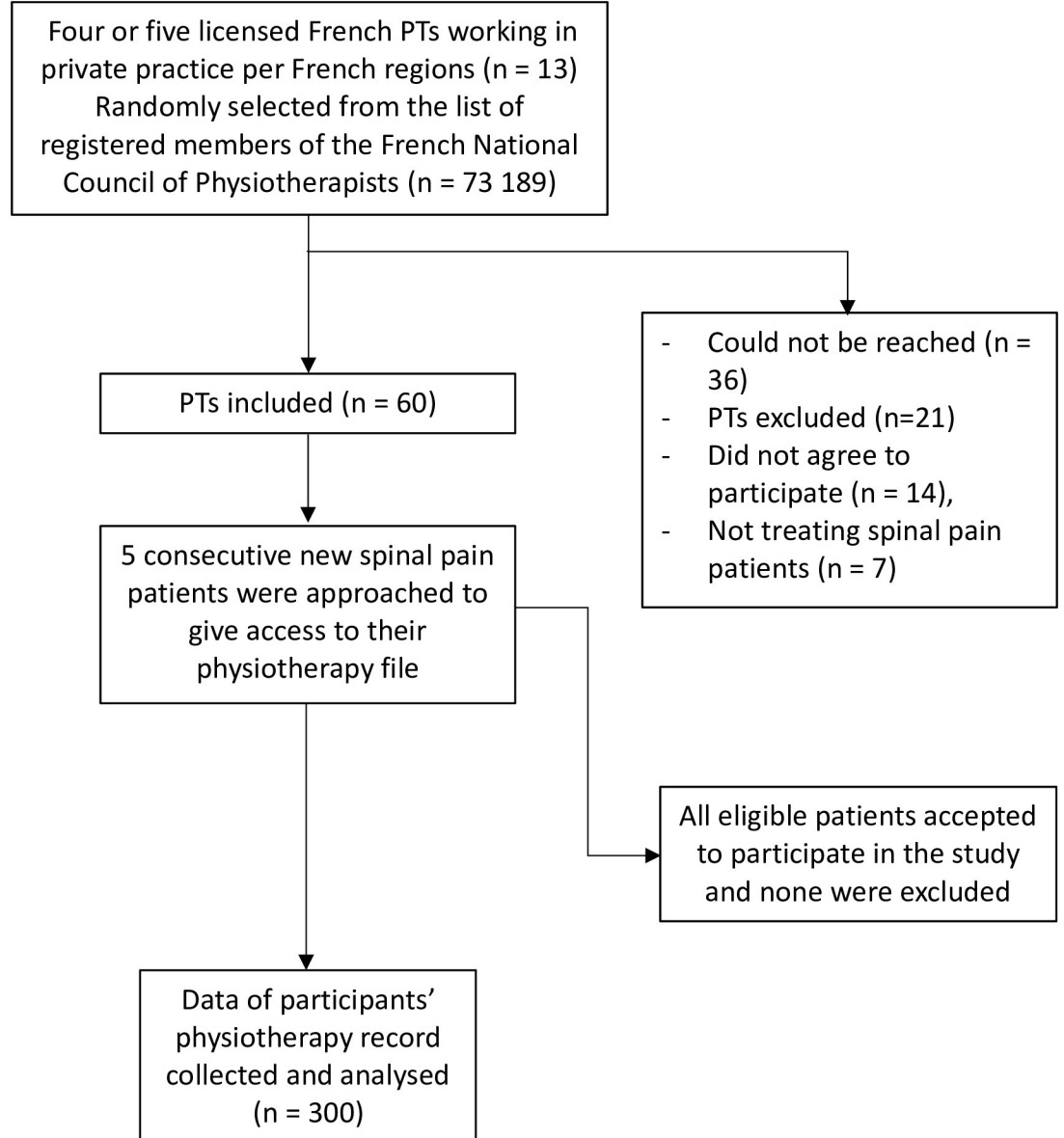

**Fig 1. Study design flowchart for inclusion of physiotherapists and spinal pain patients.** PT: physiotherapist.

referrers from the physiotherapy referral prescriptions. GPs reported the anatomical region for spinal pain complaint for 99% of all referrals (n = 297). Specific diagnoses or reason of referral formulated by GPs were present for 27% of all referred patients (n = 218). Indication of contraindications to certain physiotherapy interventions was present on 1% of referrals (n = 4). No referral included imaging or other diagnostic test results. Information on types of physiotherapy interventions prescribed by GPs were present on 54.7% of all referrals (n = 164). Specific number and frequency per week of prescribed physiotherapy consultations were present respectively on 27% (n = 80) and 8% (n = 24) of all referrals.

**Table 1. Demographic and professional characteristics of the participating French licensed physiotherapists working in private practice (n = 60).**

| Characteristics | n (%) | Mean (SD) |
|---|---|---|
| Gender | | |
| *Male* | 37 (62) | |
| *Female* | 23 (38) | |
| Age (years) | | 38.1 (10.5) |
| Clinical practice location | | |
| *Rural* | 19 (32) | |
| *Metropolitan* | 41 (68) | |
| Professional experience (years) | | 13.7 (11.1) |
| Postgraduate training in spinal rehabilitation | | |
| *Yes* | 25 (42) | |
| *No* | 35 (58) | |
| Graduating year for initial practicing PT diploma[¶] | | |
| *Between 1946 and 1989* | 6 (10) | |
| *Between 1990 and 2015* | 39 (65) | |
| Between *2016 and present* | 15 (25) | |

SD: standard deviation; PT: physiotherapy.

[¶]Categorized according to the three main reforms related to the French initial training curricula in physiotherapy (1946, 1989, and 2015).

**Table 2. Characteristics of new patients with spinal pain complaint included from the caseload of participating physiotherapists (n = 300).**

| Characteristics | n (%) | Mean (SD) |
|---|---|---|
| *Demographic characteristics* | | |
| Gender | | |
| *Male* | 140 (47) | |
| *Female* | 160 (53) | |
| Age (years) | | 48.0 (7.2) |
| Wait time between GP's referral and initial PT consultation (days) | | 12.4 (6.2) |
| *Spinal pain*[†] | | |
| Neck | 47 (16) | |
| Thoracic spine | 16 (5) | |
| Low back | 147 (49) | |
| Combination[‡] | 90 (30) | |
| *Clinical characteristics*[†] | | |
| Pain lasting more than 3 months | 142 (47) | |
| Worst spinal pain reported during initial PT consultation (VAS 0–10) | | 7.0 (2.2) |
| Presence of lower limb referred pain | 71 (24) | |
| Number of comorbidities[¶] | | 2.1 (0.9) |
| Work-related spinal pain injury | 17 (6) | |

SD: standard deviation; GP: general practitioner; PT: physiotherapist; VAS: Visual Analog Scale

[†] Reported by included patients at the initial PT consultation

[‡] Defined as concomitant neck pain, thoracic spine pain, and/or low back pain

[¶] Comorbidities of patients included were extracted from the patient's medical file and assessed by two authors (AD and LB).

**Table 3. Type of information provided by general practitioners from the physiotherapy referral of spinal pain patients (n = 300).**

| Physiotherapy referral information[†] | n (%) |
|---|---|
| Description of anatomical region for spinal pain complaint | 297 (99) |
| Presence of a specific diagnosis or reason of referral | 82 (27) |
| Indication of contraindications to certain physiotherapy interventions | 4 (1) |
| Presence of related medical information with the referral such as imaging or other diagnostic test results | 0 (0) |
| Information on types of physiotherapy interventions prescribed | 164 (55) |
| Specific number of prescribed physiotherapy consultations | 80 (27) |
| Specific prescribed frequency per week of physiotherapy consultations | 24 (8) |

[†] 164 individual general practitioners were identified from physiotherapy referral prescriptions of spinal pain patients

## Specific physiotherapy interventions prescribed by general practitioners or recommended by physiotherapists

Description and differences of specific physiotherapy interventions prescribed by GPs based on information provided on the physiotherapy referrals and recommended by PTs at their initial consultation for referred spinal pain patients (n = 164) are presented in Table 4. When specific physiotherapy interventions were prescribed by GPs, for all spinal pain patients, massage therapy was the most frequently prescribed physiotherapy intervention (78.7%, n = 129). No referral included postural and hygienic education. General exercises (whole range of motion and strengthening exercises) or stretching exercises were prescribed respectively for 28.7% (n = 47), and for 28.0% (n = 46) of all referred spinal pain patients.

For specific physiotherapy interventions recommended by PTs after their initial evaluation, specific spinal exercises, manual therapy, postural and hygienic education were respectively the three most frequently recommended interventions for all spinal pain patients (59.1%, n = 97; 59.1%, n = 97; and 39.6%, n = 65).

Compared to GPs, PTs recommended significantly more frequently specific spinal exercises (59.1%, n = 97 vs 25.0%, n = 41), manual therapy (59.1%, n = 97 vs 0.0%, n = 0), and postural and hygienic education (39.6%, n = 65 vs 0.0%, n = 0). GPs prescribed significantly more frequently passive physiotherapy approaches such as massage therapy (78.7%, n = 129 vs 37.2%, n = 61) and electrotherapy (12.2%, n = 20 vs 4.9%, n = 8).

For the 136 patients included in the study for whom no specific physiotherapy intervention was prescribed by GPs, PTs recommended similar physiotherapy interventions with specific spinal exercises, manual therapy, and postural and hygienic education being the most frequently recommended, and massage therapy, hot/cold therapy, and electrotherapy being the least frequently recommended (S4 Table in S1 File).

Overall, PTs at their initial consultation recommended significantly a lower frequency of consultation per week (mean: 1.8 consultations per week; SD ± 0.8) than GPs based on information provided on the physiotherapy referral (mean: 2.3 consultations per week; SD ± 0.6) (p = 0.02; see Table 5). There was no significant difference in the total number of physiotherapy consultations prescribed by GPs and recommended by PTs (p = 0.12, see Table 5).

## Agreement beyond chance between GPs' physiotherapy referral characteristics and PT's findings at the initial consultation

Agreements beyond chance between GPs' physiotherapy referral characteristics and physiotherapists' clinical findings at the initial consultation for spinal pain patients in terms of

**Table 4. Description and differences of specific physiotherapy interventions prescribed by general practitioners based on information provided on the physiotherapy referrals and recommended by physiotherapists at their initial consultation for referred spinal pain patients (n = 164).**

| | | Neck pain (n = 27) n (%) | Thoracic spine pain (n = 10) n (%) | Low back pain (n = 81) n (%) | Combination of spinal pain (n = 46) n (%) | All patients (n = 164) n (%) | Differences between providers P-value |
|---|---|---|---|---|---|---|---|
| Postural and hygienic education† | GPs | 0 (0.0) | 0 (0.0) | 0 (0.0) | 0 (0.0) | 0 (0.0) | *0.001 |
| | PTs | 9 (33.3) | 3 (30.0) | 30 (37.0) | 23 (50.0) | 65 (39.6) | |
| Specific spinal exercises‡ | GPs | 1 (3.7) | 1 (10.0) | 28 (34.6) | 11 (23.9) | 41 (25.0) | *0.001 |
| | PTs | 16 (59.3) | 6 (60.0) | 50 (61.7) | 25 (54.3) | 97 (59.1) | |
| McKenzie exercises | GPs | 3 (11.1) | 0 (0.0) | 0 (0.0) | 0 (0.0) | 3 (1.8) | *0.001 |
| | PTs | 5 (18.5) | 1 (10.0) | 15 (18.5) | 5 (10.1) | 26 (15.9) | |
| Stretching exercises | GPs | 3 (11.1) | 2 (20.0) | 31 (38.3) | 10 (21.7) | 46 (28.0) | 0.28 |
| | PTs | 6 (22.2) | 3 (30.0) | 30 (37.0) | 17 (37.0) | 56 (34.1) | |
| General exercises¤ | GPs | 13 (48.1) | 6 (60.0) | 18 (22.2) | 10 (21.7) | 47 (28.7) | *0.01 |
| | PTs | 3 (11.1) | 1 (10.0) | 8 (9.9) | 10 (21.7) | 22 (13.4) | |
| Manual therapy§ | GPs | 0 (0.0) | 0 (0.0) | 0 (0.0) | 0 (0.0) | 0 (0.0) | *0.001 |
| | PTs | 16 (59.3) | 6 (60.0) | 39 (48.1) | 36 (78.3) | 97 (59.1) | |
| Massage therapy | GPs | 22 (81.5) | 10 (100.0) | 66 (81.5) | 31 (67.4) | 129 (78.7) | *0.001 |
| | PTs | 9 (33.3) | 4 (40.0) | 31 (38.3) | 17 (37.0) | 61 (37.2) | |
| Hot/Cold therapy | GPs | 0 (0.0) | 0 (0.0) | 0 (0.0) | 2 (4.3) | 2 (1.2) | *0.02 |
| | PTs | 1 (3.7) | 0 (0.0) | 7 (8.6) | 3 (6.5) | 11 (6.7) | |
| Electrotherapy | GPs | 2 (7.4) | 0 (0.0) | 6 (7.4) | 12 (26.1) | 20 (12.2) | *0.03 |
| | PTs | 1 (3.7) | 1 (10.0) | 4 (4.9) | 2 (4.3) | 8 (4.9) | |
| Ultrasound therapy | GPs | 0 (0.0) | 0 (0.0) | 0 (0.0) | 3 (6.5) | 3 (1.8) | 0.25 |
| | PTs | 0 (0.0) | 0 (0.0) | 0 (0.0) | 0 (0.0) | 0 (0.0) | |

Physiotherapy interventions categories based on clinical practice guidelines and systematic reviews selected [21–24]

The physiotherapy prescription provided by the general practitioner to the patient is mandatory for the physiotherapist to be able to take care of the patient and thus have the costs covered by the French Health Insurance

* Chi-squared test or Fisher's exact test with significant value (p < 0.05) used to compare prescribed interventions prescribed/recommended to all spinal pain patients

† Such as advice on daily physical activity

‡ Defined as coordination, endurance, strengthening or postural exercises.

¤ Defined as primarily range of motion and strengthening exercise of the whole body.

§ Defined as spinal joints mobilization or manipulation and neurodynamic technique primarily tailored range of motion.

**Table 5. Comparison of specific number and frequency per week for physiotherapy consultations prescribed by general practitioners based on information provided on the physiotherapy referral and recommended by physiotherapists at their initial consultation to spinal pain patients.**

| | Mean value for GPs (SD) | Mean value for PTs (SD) | Mean difference (SD) | Mann-Whitney-Wilcoxon test | P-value |
|---|---|---|---|---|---|
| Specific number of prescribed or recommended physiotherapy consultations¶ (n = 80) | 14.0 (5.3) | 12.7 (4.5) | 1.3 (0.8) | 3650.5 | 0.12 |
| Specific prescribed or recommended frequency per week for physiotherapy consultations¶ (n = 24) | 2.3 (0.6) | 1.8 (0.8) | 0.5 (0.2) | 392.0 | *0.02 |

GPs: general practitioners; PTs: physiotherapists

¶ Categories based on the data extracted from the physiotherapy referral characteristics prescribed by GPs and recommended by PTs at their initial consultation.

According to the French law, GPs can prescribe as many sessions of physiotherapy and their frequency per week, as they deem, without any limit set by the French Health Insurance.

**Table 6.** Agreement beyond chance between general practitioners' physiotherapy referral characteristics and physiotherapists' clinical findings at the initial consultation for spinal pain patients in terms of diagnosis, contraindication to treatments, and types of physiotherapy interventions prescribed/recommended.

| | By GPs from the physiotherapy referral *n (%)* | By PTs after their initial consultation *n (%)* | Raw agreement proportion *n (%)* | Cohen's Kappa (κ) [95% CI] |
|---|---|---|---|---|
| **All specific diagnosis or reason of referral[†]** | 82/82 (100) | 82/82 (100) | 34/82 (41) | 0.19 [0.08–0.31] |
| Non-specific neck pain | 10/82 (12) | 12/82 (15) | 5/12 (42) | 0.37 [0.23–0.51] |
| Cervical radiculopathy | 3/82 (4) | 2/82 (3) | 2/2 (100) | |
| Specific neck pain[‡] | 0/82 (0) | 1/82 (1) | 0/1 (0) | |
| Non-specific thoracic spine pain | 4/82 (5) | 5/82 (6) | 4/5 (80) | 0.88 [0.63–0.95] |
| Non-specific low back pain | 20/82 (24) | 33/82 (40) | 9/33 (27) | 0.05 [0.00–0.18] |
| Radiculopathy/sciatica | 17/82 (21) | 3/82 (4) | 2/3 (67) | 0.15 [0.00–0.26] |
| Specific low back pain[ß] | 3/82 (4) | 1/82 (1) | 1/1 (100) | |
| More than one spinal pain diagnosis[¶] | 25/82 (30) | 25/82 (30) | 11/25 (44) | 0.19 [0.00–0.32] |
| **Contraindications to certain physiotherapy interventions[¤]** | 4/300 | 4/300 | 4/4 (100) | |

GP: general practitioners; PT: physiotherapists; CI: confidence interval

[†] Categories based on the diagnosis formulated by PTs after initial consultation

[‡] Whiplash (n = 1)

[ß] Spondylolisthesis (n = 1)

[¶] Agreement beyond chance obtained with all similar diagnoses of spinal pain prescribed by the GP and formulated by the PT

[¤] Pregnancy or local acute infection (n = 4 on 300 included patients)

diagnosis and contraindication to treatments are presented in Table 6. The overall proportion of agreement beyond chance for identification of a specific diagnosis or reason of referral was 41% with a weak concordance between providers (κ = 0.19; 95% CI: 0.08–0.31). For specific spinal diagnoses, proportions of agreement beyond chance varied from weak to very high, such as for non-specific neck pain (κ = 0.37; 95% CI: 0.23–0.51), non-specific thoracic spine pain (κ = 0.88; 95% CI: 0.63–0.95), non-specific low back pain (κ = 0.05; 95% CI: 0.00–0.18), radiculopathy/sciatica (κ = 0.15; 95% CI: 0.00–0.26), and for patients with multiple spinal pain diagnoses (κ = 0.19; 95% CI: 0.00–0.32). The overall proportion of agreement beyond chance for contraindications for certain physiotherapy techniques was 100% (reported for 4 participants in our study).

## Proportions of GPs and PTs conforming to recommendations for spinal pain care

Based on all spinal pain patients, 47.0% of GPs (77/164) and 76.7% of PTs (46/60) prescribed at least one physiotherapy intervention supported by moderate to strong evidence (Table 7). The proportion of compliant PTs was significantly higher than for GPs when comparing all spinal pain patients (76.7% vs 47.0%; p<0.001). Other analyses based on each type of spinal pain patients reported that the proportion of compliant PTs was significantly higher compared to GPs for the physiotherapy treatment plan for low back pain patients (87.5% vs 40.7%; p<0.001), but not for other spinal pain patients categories, although a tendency was observed in favor of PTs compared to GPs (respectively for neck pain patients, 77.8% vs 48.1%, p = 0.24; thoracic spine pain patients, 80.0% vs 40.0%, p = 0.28; and combination of spinal pain, 63.6% vs 58.7%, p = 0.70).

**Table 7. Proportion of general practitioners (n = 164) and physiotherapists (n = 60) prescribing at least one physiotherapy intervention for spinal patients based on recommendations from clinical practice guidelines for the management of spinal pain patients.**

|  | General practitioners<br>n (%) | Physiotherapists[†]<br>n (%) | P-value |
|---|---|---|---|
| **All spinal pain patients** | 77/164[‡] (47.0) | 46/60 (76.7) | *0.001 |
| Neck pain patients | 13/27 (48.1) | 7/9 (77.8) | 0.24 |
| Thoracic spine pain patients | 4/10 (40.0) | 4/5 (80.0) | 0.28 |
| Low back pain patients | 33/81 (40.7) | 21/24 (87.5) | *0.001 |
| Combination of spinal pain[¶] | 27/46 (58.7) | 14/22 (63.6) | 0.70 |

[†] Each of the physiotherapists can be found in one or more categories of spinal pain patients depending on the spinal pain area of the five included patients seen at consultation

[‡] 164 individual general practitioners were identified from physiotherapy referral prescriptions of spinal pain patients

*Chi-squared test or Fisher's exact test with significant value (p < 0.05)

[¶] Defined as concomitant neck pain, thoracic spine pain, and/or low back pain

## Discussion

The aims of this study were first to describe types of spinal pain patients referred by their treating GP to physiotherapy care based on information from the GPs' referral form, then to examine to which extent specific physiotherapy interventions, prescribed by GPs and PTs, were adherent to evidence based recommendations and finally to evaluate concordance between GPs' physiotherapy referral characteristics and PTs' findings at their initial consultation. Sixty PTs from the 13 geographical regions of France were recruited and, for each PT, five consecutive patients newly referred for spinal pain complaints by their GP were included for a total of 300 participants.

Among the sample of 300 spinal pain patients included, low back pain was the most frequently reported complaint. For all types of spinal pain, almost half had a chronic condition and they presented severe pain intensity with a mean of 7 out of 10 (SD ± 2.2). For the vast majority of patients referred by GPs to PTs, several important information needed by the PT, such as a complete diagnosis or reason of referral, imaging exam or other diagnostic test results, were not provided. The lack of relevant information provided by GPs suggests that PTs in this GP-led model of care may not have all important information to care safely for these patients. Thus, PTs need also to make a diagnosis based on detailed history and clinical findings and may need, if the patient condition requires it (e.g. if a serious condition or even if a red flag is suspected), to refer back the patient to the GP if physiotherapy is not indicated and this could delay care.

For spinal pain patients for whom a complete diagnosis or reason of referral had been formulated by GPs and presented on the referral form, the diagnostic concordance with PTs was only weak. The lack of agreement beyond chance between GPs and PTs could contribute to suboptimal care and treatment for the patient's condition [26, 27]. Although we cannot conclude for certain with the present results that GPs diagnoses were inadequate. Based on available evidence, nonspecific diagnoses are frequently provided by GPs with limited referral information to PTs [26]. The ability of non-specialized musculoskeletal trained health professionals such as GPs to formulate an accurate diagnosis is often questioned in the literature [26, 27]. When compared to two references such as orthopaedic surgeons' diagnosis findings or magnetic resonance imaging findings, clinical diagnoses from PTs had a significant higher accuracy compared to those from GPs [27].

In terms of physiotherapy treatments, specific physiotherapy interventions and the overall number of consultations prescribed by GPs were however reported for several patients. Massage therapy was the most frequently prescribed specific physiotherapy intervention by GPs. Exercises, postural and hygienic education were the least prescribed interventions. This is concerning as these treatments are considered now low-quality treatments and could lead to poorer outcomes or chronicization of the patients' condition. Selected CPGs and systematic reviews strongly recommend the use of active therapeutic approaches such as exercises and limit passive approaches such as massage therapy, stretching exercises, and physical modalities for the management of spinal pain patients [21–24]. For the participating PTs, exercises, manual therapy, or postural and hygienic education were respectively the three most frequently recommended specific physiotherapy interventions for all spinal pain patients. The overall number of consultations prescribed by GPs or recommended by PTs was not significantly different between providers, but in terms of prescribed physiotherapy consultations per week GPs prescribed significantly more sessions per week than PTs. Our results appear consistent with the findings of several studies evaluating practice patterns of GPs compared to PTs in other countries [8, 14, 15, 17, 18, 28, 29]. Based on these studies' results, authors reported low compliance of GPs' when prescribing physiotherapy and from low to high compliance of PTs to evidence based recommendations for physiotherapy care of spinal pain patients. From our findings, less than half of GPs prescribed physiotherapy care in compliance with recommendations, while three-quarters of PTs did so for the physiotherapy care of these patients. However, it should be noted that almost 25% of the PTs were not compliant with recommendations. A non-optimal physiotherapy referral provided by the GP could lead the PT to follow the prescribed physiotherapy interventions, even if such interventions are not evidence based. A GP's prescription containing specific indications regarding the physiotherapy care to be delivered (such as type of interventions, number and/or frequency per week of physiotherapy consultations) can have a strong impact on the patient's expectations. This can jeopardize the patient's trust in the PT who wants to plan the most appropriate treatment for the patient's condition, which might not be the one recommended by the GP [30]. The consequences may be delay in the patient's recovery due to inadequate quality of care and potential clinical complications resulting in increased health care costs [31]. Thus, it might be suggested that a majority of patients seeking care for spinal pain could benefit directly from the services offered by a PT providing care in compliance with evidence based recommendations [32, 33]. French PTs undergo extensive training and have specialized skills to assess and treat spinal pain patients [34–36] according to the latest reform of the initial training of French licensed PTs in 2015 [37]. Due to the frequent presentation in primary care of patients consulting with spinal pain complaint, most of which is considered benign and directly indicated to physiotherapy care [38], it is essential to question the relevance of the primary care model led by GPs and potentially allow PTs to offer their services directly to these patients without prior referral.

## Strengths and limitations

This is the first study conducted in France presenting results regarding which patients and how spinal pain patients are referred by GPs to PTs in French GP-led primary care and compliance of the physiotherapy care proposed by GPs and PTs to these patients as regards to CPGs and systematic reviews. An important number of spinal pain patients physiotherapy records was included (n = 300) from sixty participating PT across France. The analysis of the information given by the GP to the physiotherapist through their referral allows to estimate the real life work and sometimes difficulties encountered by physiotherapists. This study

reports the usual practices of French physiotherapists with the only patients they actually encountered: those referred by their GP, and the way they are referred.

Our study presents some methodological limitations. One of the main limitation is that we used information from GPs' physiotherapy referral prescribed may not fully reflect the GP's practices as they were not directly surveyed specifically about their patient's diagnoses, the reason for referring the patient to physiotherapy, and medical information from imaging or other diagnostic test results. Yet, this study reflects real life communication between those two healthcare providers and then their practices within the French context. Furthermore, the author's choice to define the perfect overall raw agreement between providers for specific physiotherapy interventions prescribed or recommended by providers based on the presence of at least one concordant physiotherapy intervention could be discussed, although clinical practice guidelines recommend more than one specific physiotherapy intervention for the management of spinal pain patients. This choice may therefore represent the most optimistic scenario for assessing inter-rater agreement, as it does not allow for an assessment of the heterogeneity of the providers' practices prescribing or recommending more than two specific interventions beyond the defined agreement.

The low participation rate (43%) from eligible PTs may reflect a nonresponse bias, because the respondents were probably more motivated to participate to this survey because they were potentially more up to date on the management of spinal pain than were nonresponders, however the majority did not have specific spinal pain training. Our study might present an observer bias from participating PTs because, knowing the research aims and their inclusion in the study, they were able to take more time during their initial consultation to potentially better manage the spinal pain patients included. The selection of recent evidence-based recommendations of CPGs and systematic reviews published for the management of adults with spinal pain was not conducted from a systematic electronic literature search to ensure that specific physiotherapy interventions classifications used for spinal pain patients agreed upon in the scientific literature, but the recommendations used in our study are in line with several other CPGs or reviews [39–42].

## Conclusions

In this cross-sectional study of French spinal pain patients, we found that information required for the referral of these patients to physiotherapy is often incomplete. The majority of GPs did not conform to evidence-based recommendations in terms of prescribed specific physiotherapy care. Considering that MSKDs are encountered by GPs more and more frequently, it would be interesting to develop and disseminate simple diagnostic and referral decision trees, built jointly by GPs, MSKDs specialized physicians and PTs. This would allow a common language between GPs and PTs, and a better fluidity of care for patients. In parallel, it would be interesting to explore through safety and effectiveness studies models increasing PTs autonomy for patients seeking care for spinal pain.

## Supporting information

**S1 File.**
(ZIP)

## Acknowledgments

We would like to thank all the physiotherapists and patients who participated in this study.

## Author Contributions

**Conceptualization:** Anthony Demont, François Desmeules, Aurélie Bourmaud.

**Data curation:** Anthony Demont, Aurélie Bourmaud.

**Formal analysis:** Anthony Demont, François Desmeules, Aurélie Bourmaud.

**Funding acquisition:** Anthony Demont, Aurélie Bourmaud.

**Investigation:** Anthony Demont, Leila Benaïssa, Valentine Recoque.

**Methodology:** Anthony Demont, François Desmeules, Aurélie Bourmaud.

**Project administration:** Anthony Demont, Aurélie Bourmaud.

**Resources:** Anthony Demont, Leila Benaïssa, Valentine Recoque.

**Supervision:** François Desmeules, Aurélie Bourmaud.

**Validation:** François Desmeules, Aurélie Bourmaud.

**Writing – original draft:** Anthony Demont.

**Writing – review & editing:** Anthony Demont, Leila Benaïssa, Valentine Recoque, François Desmeules, Aurélie Bourmaud.

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
