## [Decision Letter · Decision Letter 0]

17 May 2022

PONE-D-21-38018

Spinal pain patients seeking care in primary care and referred to physiotherapy: A cross-sectional study on patients characteristics, referral information and physiotherapy care offered by general practitioners and physiotherapists in France

PLOS ONE

Dear Dr. DEMONT,

Thank you for submitting your manuscript to PLOS ONE. After careful consideration, we have decided that your manuscript does not meet our criteria for publication and must therefore be rejected.

Both reviewers suggested some concerns. However, reviewer 1 indicated a major issues. Thus, your manuscript was rejected based on these comments.

I am sorry that we cannot be more positive on this occasion, but hope that you appreciate the reasons for this decision.

Kind regards,

Fatih Özden, PhD

Academic Editor

PLOS ONE

Reviewers' comments:

Reviewer's Responses to Questions

**Comments to the Author**

1. Is the manuscript technically sound, and do the data support the conclusions?

Reviewer #1: Partly

Reviewer #2: Yes

2. Has the statistical analysis been performed appropriately and rigorously? 

Reviewer #1: I Don't Know

Reviewer #2: Yes

3. Have the authors made all data underlying the findings in their manuscript fully available?

Reviewer #1: Yes

Reviewer #2: Yes

4. Is the manuscript presented in an intelligible fashion and written in standard English?

Reviewer #1: Yes

Reviewer #2: Yes

5. Review Comments to the Author

Reviewer #1: Line 35. The sentence „the purpose of this referral is to validate the indication for physiotherapy care, to exclude potential red flags” is misleading. It is not appropriate to refer patients with suspected red flag pathologies to physiotherapists. I assume you agree. They should rather have imaging or be referred to a specialist, depending on the urgency and severity of symptoms.

Line 40. I disagree that reference 3 is reporting under-referral to physiotherapy. It is a descriptive study. I even doubt that over and under-referral can reasonably be assessed. Guidelines a very unspecific about which type of physiotherapy should be used and how to select appropriate patients. Physiotherapy takes time and is difficult to fit in the schedule of working people, therefor some patients do The issue of overprescribing physiotherapy is not addressed. For acute pain PT is not recommended. On the other hand GPs are often pressured to prescribe massage, which is considered of limited value.

Given that patients are recruited in physiotherapy and not in Genera Practice the questions of potential under-referral cannot be addressed with this study. Therfor this section of the introduction is not pertinent for the manuscript.

Overall the rational for describing the characteristics of patients referred from GPs to physiotherapy is weak, given that the characteristics cannot be compared to those who didn’t receive a referral. Information on any other medical interventions is missing.

Line 64. The STROBE statement is mentioned but the checklist is not available in the supplement.

Line 67. Setting. International readers need a little bit more elaboration. In some countries patients have direct access to PT. Do patients need a referral? Is there a co-payment?

If patients need a referral who is ultimately selecting which type physiotherapy is delivered the prescriber or the PT? Is prescription really so specific, in many countries PT assess patients and select themselves the therapy. This is important to understand why you are comparing GPs and PTs agreement.

Which information on the patient is provided by the GP on the referral? Is there a standardized referral form (see also line 96 to 106)

Line 126. A specific diagnosis can often not be established for musculoskeletal pain. Therefor LBP is mostly classified as unspecific. Most diagnosis are descriptive. Probably the term working diagnosis is more appropriate.

Line 132-44. The criteria for assessing guideline concordance with referral to physiotherapy are not clearly stated. The list S1 in the appendix does not help to understand what exactly was considered in accordance with the guideline. I am not aware of any guideline being very specific regarding exercise therapy.

How was education and reassurance provided by GPs recorded?

The problem is even worse. It is reasonable to expect that GPs stick to a national guideline. However, expecting them to stick to various international guidelines seems inappropriate. Is there a French guideline which considered as authoritative for French GPs?

It seems you are assessing use of evidence based PT and not compliance or adherence to guidelines.

Line 207-214. See question above about how physiotherapy is prescribed in France. In some countries the number of PT sessions is tightly regulated. This need to be explained. Can GPs prescribe any number of sessions? Can PT influence the number of sessions? Table 5 needs more background information.

It seems awkward that you are mentioning prescription of reassurance and education. First of all is reassurance and education something which can be prescribed. Isn’t it regarded as part of any intervention delivered in PT? Is reassurance not the responsibility of the GPs?

Table 4 is difficult to interpret without better understanding how PT is prescribed an billed.

Line 245-254. PT with training in manual therapy will always make a very specific functional diagnosis unlike GPs without training in manual therapy. Without information how diagnosis are coded, e.g. ICD-10 and if GPs and PT use the same codes or if the can pick freely the diagnosis the rationale for comparing agreement between GPs and PTs seems questionable.

Line 265. Again, it is puzzling. Who is prescribing the GP or the PT?

The discussion of the findings is week. Many factors which seems to be related to the French health care system (hard to say since it is hard to understand how PT is prescribed and delivered from the available information).

Limitations are mentioned

The conclusion has no implications for practice or research.

Reviewer #2: Comments

This manuscript reports the analysis of a cross-sectional multicentered observational study to evaluate aspects related to referral and proposed interventions by general practitioners and physiotherapists for spinal pain patients. The background and rationale of the study are very well posed, and the study aims are clearly stated. The study is reported following appropriated guidelines (STROBE). The use of a random sampling scheme at national level along with a standard questionnaire developed by a multidisciplinary team are major strengths of the study. Data analysis is properly conducted. The findings are interesting as they highlight a poor agreement beyond chance between general practitioners and physiotherapists regarding spinal pain diagnosis; and the overall low (high) evidence-based treatment referrals by general practitioners (physiotherapists) as compared to current guidelines. I commend the authors for executing and reporting a high-quality study; I have only minor suggestions for their considerations.

Minor comments

1. In additional to the STROBE, consider double-checking if your manuscript includes all relevant information for reporting agreement and reliability studies (GRRAS; https://www.equator-network.org/reporting-guidelines/guidelines-for-reporting-reliability-and-agreement-studies-grras-were-proposed/).

2. When mentioning estimates of agreement using kappa, consider mentioning that is assessed ‘agreement beyond chance’, in contrast to absolute and relative frequencies that are no adjusted to chance.

3. As a methodological choice, the authors measured the overall raw agreement between providers for specific physiotherapy interventions prescribed or recommended by providers defining that the presence of at least two concordant physiotherapy interventions was a perfect agreement. I suggest discussing this choice as this seems the more optimistic scenario for assessing interrater agreement. Also, it may be interpreted as a ‘believe the positive’ strategy, whereas other strategies (e.g., ‘believe the negative’) could be used.

6. PLOS authors have the option to publish the peer review history of their article (what does this mean?). If published, this will include your full peer review and any attached files.

Reviewer #1: **Yes: **Jean-Francois Chenot

Reviewer #2: **Yes: **Arthur de Sá Ferreira

- - - - -

---

## [Author Response · Author response to Decision Letter 0]

20 Jun 2022

Response to comments from the editor and reviewers - Manuscript PONE-D-21-38018

We appreciate the time invested by the Editor and reviewers for their thoughtful comments and for the opportunity given to improve our manuscript for consideration. We provide below the detailed responses to comments.

Associate Editor:

1. In additional to the STROBE, consider double-checking if your manuscript includes all relevant information for reporting agreement and reliability studies (GRRAS; https://www.equator-network.org/reporting-guidelines/guidelines-for-reporting-reliability-and-agreement-studies-grras-were-proposed/).

Thank you for this suggestion. We have added the conformity to the GRRAS in the Method section (p4, line 65) and added the GRRAS checklist completed in the Supporting information (S4 Appendix).

2. When mentioning estimates of agreement using kappa, consider mentioning that is assessed ‘agreement beyond chance’, in contrast to absolute and relative frequencies that are no adjusted to chance.

Correction was done (Abstract section: p2, line 18; Method section: p8, lines 171 and 182; Results section: p20, lines 267,269,271,273,276; p21, lines 278,285; Discussion section: p24, line 320).

3. As a methodological choice, the authors measured the overall raw agreement between providers for specific physiotherapy interventions prescribed or recommended by providers defining that the presence of at least two concordant physiotherapy interventions was a perfect agreement. I suggest discussing this choice as this seems the more optimistic scenario for assessing interrater agreement. Also, it may be interpreted as a ‘believe the positive’ strategy, whereas other strategies (e.g., ‘believe the negative’) could be used.

We have added this issue in the Discussion section (p26, lines 372-378), which now quotes : “Furthermore, the author’s choice to define the perfect overall raw agreement between providers for specific physiotherapy interventions prescribed or recommended by providers based on the presence of at least one concordant physiotherapy intervention could be discussed, although clinical practice guidelines recommend more than one specific physiotherapy intervention for the management of spinal pain patients. This choice may therefore represent the most optimistic scenario for assessing inter-rater agreement, as it does not allow for an assessment of the heterogeneity of the providers’ practices prescribing or recommending more than two specific interventions beyond the defined agreement.“

Reviewer #1: 

1. Line 35. The sentence „the purpose of this referral is to validate the indication for physiotherapy care, to exclude potential red flags” is misleading. It is not appropriate to refer patients with suspected red flag pathologies to physiotherapists. I assume you agree. They should rather have imaging or be referred to a specialist, depending on the urgency and severity of symptoms.

Correction was done, it was indeed a mistake (p3, line 36).

2. Line 40. I disagree that reference 3 is reporting under-referral to physiotherapy. It is a descriptive study. I even doubt that over and under-referral can reasonably be assessed. Guidelines a very unspecific about which type of physiotherapy should be used and how to select appropriate patients. Physiotherapy takes time and is difficult to fit in the schedule of working people, therefor some patients do The issue of overprescribing physiotherapy is not addressed. For acute pain PT is not recommended. On the other hand GPs are often pressured to prescribe massage, which is considered of limited value. Given that patients are recruited in physiotherapy and not in General Practice the questions of potential under-referral cannot be addressed with this study. Therefore this section of the introduction is not pertinent for the manuscript.

It is possible that our reviewer made a reading error here. Indeed, reference 3 does not call for a descriptive study from which the problem of under-referencing could not be addressed. But reference 3 is a systematic review analyzing several randomized controlled studies reporting data on the contents and outcomes of the usual treatments offered in primary care by general practitioners to patients with low back pain. The included studies reported that there was a high frequency of patient referrals by GPs for medication and conversely weak evidence suggesting that they promoted physical activity. 

However, in order to avoid any misunderstanding, or any impression of hasty judgment on our part, we modified the sentence according with the exact content of this systematic review and the other studies cited (p3, lines 41-42). This section now quotes “poor patient education as well as poor promotion of active treatments such as physical activity is often reported”.

3. Overall the rationale for describing the characteristics of patients referred from GPs to physiotherapy is weak, given that the characteristics cannot be compared to those who didn’t receive a referral. Information on any other medical interventions is missing.

We agree with the reviewer that our study couldn’t describe characteristics of patients not referred to physiotherapy. However, according to the objectives of this study, the design of this study was adapted and valid to address all the issues raised by those objectives, which were not those reported above. 

The aims of the study were: 1- to describe types of spinal pain patients referred by their treating GPs to participating PTs, based on information collected from the GPs’ physiotherapy referral form; 2- based on information on the GPs’ physiotherapy referral, to examine to which extent, when specific physiotherapy interventions are prescribed by GPs, they adhere to evidence-based recommendations for care of these patients; 3- based on information in the patient’s physiotherapy record, to examine to which extent physiotherapy interventions provided by the treating PT adhere to evidence-based recommendations for care of these spinal pain patients, and 4- to compare and evaluate concordance between information provided by the GP from the physiotherapy referral and the treating PT after their initial consultation on diagnosis and prescribed physiotherapy interventions. Thus, this study and its results are centered on patients referred to physiotherapy. 

Furthermore, those data (extracted from general practitioners’ prescriptions), based on the few information the physiotherapist gets at their initial consultation for referred spinal pain patients, reflect the characteristics usually transmitted in real life. Which represents exactly the objectives of this study, which is to describe real life practices and their potential impact on the practices of physiotherapists. 

What’s more the presence of medical information that could influence physiotherapy care was collected such as indication of contraindications to certain physiotherapy interventions and presence of related medical information provided with the referral such as imaging or other diagnostic test results as shown in Table 3 (see page 14).

In order to clarify our position regarding the objectives centered on patients referred to physiotherapy, we added a sentence in the discussion section (p26, lines 364-367) :

“The analysis of the information given by the GP to the physiotherapist through their referral allows to estimate the real life work and sometimes difficulties encountered by physiotherapists. This study reports the usual practices of French physiotherapists with the only patients they actually encountered: those referred by their GP, and the way they are referred.”.

4. Line 64. The STROBE statement is mentioned but the checklist is not available in the supplement.

We added the completed STROBE checklist in Supporting information (S3 Appendix). Furthermore, as suggested by the Editor, we have added the completed GRRAS checklist to the Supporting information as well (S4 Appendix).

5. Line 67. Setting. International readers need a little bit more elaboration. In some countries patients have direct access to PT. Do patients need a referral? Is there a co-payment?

Clarifications have been added (p4, lines 70-72). This section now quotes: “Based on the French law, patients with spinal pain seeking physiotherapy care cannot access a PT directly. They require a prescription from a physician to refer to a PT whose care will be covered by the French National Health Insurance.”

1. If patients need a referral who is ultimately selecting which type physiotherapy is delivered the prescriber or the PT? Is prescription really so specific, in many countries PT assess patients and select themselves the therapy. This is important to understand why you are comparing GPs and PTs agreement.

We agree with the reviewer that this issue deserves to be more clearly stated. Indeed physiotherapists, as care providers, are the health professionals who ultimately choose what specific physiotherapy interventions to provide to spinal pain patients. However, patients are sensitive to the indications provided on the GP’s prescription which they may take as recommendations of « what is appropriate to receive as physiotherapy care ». Thus, this is what is pointed in the Discussion (p25, lines 344-346), two cases can co-exist : 

1) the first with physiotherapists following the indications provided on the GP’s prescription even if these are not based on clinical practice guidelines (to avoid confronting the patient's expectation that the treatment provided is the same as the one described on the prescription), 

2) the second with physiotherapists not following the indications provided on the prescription by the physician and having to justify this to the patient.

We added a specific point regarding the effects of GP prescriptions on patients' expectations of physiotherapy care in the discussion section (p25, lines 346-350), which now quotes : “A GP's prescription containing specific indications regarding the physiotherapy care to be delivered (such as type of interventions, number and/or frequency per week of physiotherapy consultations) can have a strong impact on the patient's expectations. This can jeopardize the patient’s trust in the PT who wants to plan the most appropriate treatment for the patient’s condition, which might not be the one recommended by the GP.”

2. Which information on the patient is provided by the GP on the referral? Is there a standardized referral form (see also line 96 to 106)

We are sorry for this lack of precision. Clarification was added (p6, lines 113-114). This section now quotes : “from the standardized prescription form used in clinical practice, when these characteristics and information have to be reported:”

3. Line 126. A specific diagnosis can often not be established for musculoskeletal pain. Therefor LBP is mostly classified as unspecific. Most diagnosis are descriptive. Probably the term working diagnosis is more appropriate.

We are sorry for this wording issue. Correction was done. The text now quotes (p6, line 130 and p8, line 169): “specific working diagnosis”.

4. Line 132-44. The criteria for assessing guideline concordance with referral to physiotherapy are not clearly stated. The list S1 in the appendix does not help to understand what exactly was considered in accordance with the guideline. I am not aware of any guideline being very specific regarding exercise therapy.

We are sorry for this insufficient description. As regards to the selection criteria for coding concordance, we completed the list S1 with our assessing process. 

As regards to the guidelines, we also added additional information : the materials and methods section now quotes (p7, lines 146-155): “In the absence of recent French CPGs for the management of patients with neck or thoracic spinal pain, we chose international guidelines (22-24), based on the highest level of scientific evidence from various competent authorities recognized internationally for the quality of their scientific productions. These CPGs are not specific to physiotherapists but to all primary care health professionals taking care of these populations of patients. In France, GPs must keep informed of the latest published medical evidence in order to adapt their practices, through professional development. This is a requirement of best medical practice for all medical doctors. The different specific physiotherapy interventions prescribed by GPs and recommended by PTs were categorized in a standardized manner by the authors (AD and LB) according to the most appropriate category of physiotherapy interventions from the selected CPGs; a third evaluator was involved if consensus on the most appropriate physiotherapy intervention category was not reached (AB) (S1 Appendix).”

5. How was education and reassurance provided by GPs recorded?

We are sorry, we expressed ourselves poorly. “Education and reassurance” refer to postural and hygenic education or advices, such as postural hygiene or advice on daily physical activity. They refer to a category of interventions that could be delivered by physiotherapists. Those were prescribed by GPs to be delivered by physiotherapists. Thus, it does not refer to education delivered by the physician himself, which could not be evaluated in this study. In order to improve understanding, we changed “education and reassurance” by “postural and hygenic education” (Method section: p6, line 121; Results section: p15, line 230; Table 4; p17, lines 241,246,249,253-254; and Discussion section: p25, lines 330,335).

6. The problem is even worse. It is reasonable to expect that GPs stick to a national guideline. However, expecting them to stick to various international guidelines seems inappropriate. Is there a French guideline which considered as authoritative for French GPs? It seems you are assessing use of evidence based PT and not compliance or adherence to guidelines.

We are sorry, we were not clear on the description of the guidelines. This has been discussed earlier in the review. 

The text now quotes (p7, lines 146-155): “In the absence of recent French CPGs for the management of patients with neck or thoracic spinal pain, we chose international guidelines (22-24), based on the highest level of scientific evidence from various competent authorities recognized internationally for the quality of their scientific productions. These clinical practice guidelines are not specific to physiotherapists but to all primary care health professionals taking care of these populations of patients. In France, GPs must keep informed of the latest published medical evidence in order to adapt their practices, through professional development. This is a requirement of best medical practice for all medical doctors.“ 

The S1 list has been explained accordingly.

Among the different specific physiotherapy interventions of spinal pain in the included patients' records, recent French clinical practice guidelines published by the French National Health Authority exist for the management of low back pain (published by Bailly et al, 2021 and on the FNAH website in french), to which general practitioners must conform (population targeted by the recommendations as indicated in the rationale). For neck pain (Parikh et al, 2019 and Babatunde et al, 2017), thoracic spine pain (Southerst et al, 2015 and Babatunde et al, 2017) and combination of spinal pain (defined as neck pain, thoracic spine pain, and/or low back pain) (Babatunde et al, 2017), we chose international guidelines, based on the highest level of scientific evidence from various competent authorities recognized internationally for the quality of their scientific productions, due to the absence of recent French guidelines. These clinical practice guidelines are not specific to physiotherapists but to all primary care health professionals taking care of these populations of patients. 

In France, when there are no recommendations from scientific societies for a specific condition, which is often the case, doctors have a duty of professional development: they must keep informed of the latest published medical evidence in order to adapt their practices. Thus, they have an obligation to respect the results of the latest reviews or articles of the highest level of evidence, for the management of their patients. This is a requirement of best medical practice in our country, and this is why we have been able to select a panel of recommendations, when these existed, and high level evidence reviews to determine, in accordance with our French best medical practice, our list of recommendations. 

7. Line 207-214. See question above about how physiotherapy is prescribed in France. In some countries the number of PT sessions is tightly regulated. This need to be explained. Can GPs prescribe any number of sessions? Can PT influence the number of sessions? Table 5 needs more background information.

We are sorry that we did not describe the French situation enough. This has been corrected in the text which now says (p4, lines 70-72): “Based on the French law, patients with spinal pain seeking physiotherapy care cannot access a PT directly. They require a prescription from a physician to refer to a PT whose care will be covered by the French National Health Insurance” . 

Likewise, there is no limitation in the French law for the required number of sessions needed (they are not subject to a reference framework by the French Health Insurance). 

Clarifications have been added accordingly in Table 5 (p19, lines 265-266), “According to the French law, GPs can prescribe as many sessions of physiotherapy and their frequency per week, as they deem, without any limit set by the French Health Insurance”. 

8. It seems awkward that you are mentioning prescription of reassurance and education. First of all is reassurance and education something which can be prescribed. Isn’t it regarded as part of any intervention delivered in PT? Is reassurance not the responsibility of the GPs?

Once again, we are sorry we expressed ourselves poorly. “Education and reassurance” refers to postural and hygienic education or advices, such as postural hygiene or advice on daily physical activity. They refer to a category of interventions that are delivered by physiotherapists and not GPs in such cases. 

In order to improve understanding, we changed “education and reassurance” by “postural and hygenic education” (Method section: p6, line 121; Results section: p15, line 230; Table 4; p17, lines 241,246,249,253-254; and Discussion section: p25, lines 330,335).

9. Table 4 is difficult to interpret without better understanding how PT is prescribed an billed.

As mentioned above, we are sorry for not having describe sooner the French system. The text now quotes (Table 4, p16, lines 237-238) : “The physiotherapy prescription provided by the general practitioner to the patient is mandatory for the physiotherapist to be able to take care of the patient and thus have the costs covered by the French Health Insurance”.

10. Line 245-254. PT with training in manual therapy will always make a very specific functional diagnosis unlike GPs without training in manual therapy. Without information how diagnosis are coded, e.g. ICD-10 and if GPs and PT use the same codes or if the can pick freely the diagnosis the rationale for comparing agreement between GPs and PTs seems questionable.

We understand the reviewer concern. This is why we controlled for this bias in the study design.

In order to clarify this, we added in the method section (p8, lines 175-178), the coding procedure performed in this study : “Generic diagnostic coding was performed by two independent authors (AD and VR) both for GPs and PTs, from the diagnoses reported by each of these providers when present. This was to avoid ontological differences as well as medical versus working specificities. The objective was to ensure that diagnoses provided by GPs and PTs were comparable”.

11. Line 265. Again, it is puzzling. Who is prescribing the GP or the PT?

The discussion of the findings is week. Many factors which seems to be related to the French health care system (hard to say since it is hard to understand how PT is prescribed and delivered from the available information).

We have modified the Method and Results sections to clarify how the prescribing of specific physiotherapy interventions by general practitioners and the delivery of these interventions by physiotherapists is formalized under the French Health Insurance (p4, lines 70-72; Table 4, p17, lines 237-238; and Table 5, p19, lines 265-266). In order to underline that those results are in part related to the French Health system organization, we added in the limitation section (p26, lines 371-372) : “Yet, this study reflects real life communication between those two healthcare providers and then their practices within the French context“.

12. The conclusion has no implications for practice or research.

We are sorry that the conclusion failed to draw out clear implications for practice and research. The conclusion now quotes (p27, lines 393-397): “Considering that MSKDs are encountered by GPs more and more frequently, it would be interesting to develop and disseminate simple diagnostic and referral decision trees, built jointly by GPs, MSKDs specialized physicians and PTs. This would allow a common language between GPs and PTs, and a better fluidity of care for patients. In parallel, it would be interesting to explore through safety and effectiveness studies models increasing PTs autonomy for patients seeking care for spinal pain.”

Reviewer #2: 

1. This manuscript reports the analysis of a cross-sectional multicentered observational study to evaluate aspects related to referral and proposed interventions by general practitioners and physiotherapists for spinal pain patients. The background and rationale of the study are very well posed, and the study aims are clearly stated. The study is reported following appropriated guidelines (STROBE). The use of a random sampling scheme at national level along with a standard questionnaire developed by a multidisciplinary team are major strengths of the study. Data analysis is properly conducted. The findings are interesting as they highlight a poor agreement beyond chance between general practitioners and physiotherapists regarding spinal pain diagnosis; and the overall low (high) evidence-based treatment referrals by general practitioners (physiotherapists) as compared to current guidelines. I commend the authors for executing and reporting a high-quality study; I have only minor suggestions for their considerations.

The authors thank the reviewer #2 for these positive comments. 

We have added clarifications regarding estimates of agreement using kappa with « agreement beyond chance » (Abstract section: p2, line 18; Method section: p8, lines 171 and 182; Results section: p20, lines 267,269,271,273,276; p21, lines 278,285; Discussion section: p24, line 320).

---

## [Decision Letter · Decision Letter 1]

28 Jul 2022

PONE-D-21-38018R1Spinal pain patients seeking care in primary care and referred to physiotherapy: A cross-sectional study on patients characteristics, referral information and physiotherapy care offered by general practitioners and physiotherapists in FrancePLOS ONE

Dear Dr. DEMONT,

Thank you for submitting your manuscript to PLOS ONE. After careful consideration, we feel that it has merit but does not fully meet PLOS ONE’s publication criteria as it currently stands. Therefore, we invite you to submit a revised version of the manuscript that addresses the points raised during the review process.

We look forward to receiving your revised manuscript.

Kind regards,

Walid Kamal Abdelbasset, Ph.D.

Academic Editor

PLOS ONE

Journal Requirements:

Additional Editor Comments (if provided):

1. The rationale of the study should be explained in detail. You should focus on three elements of introduction:

a. What is known about the topic? (Background)

b. What is not known? (The research problem)

c. Why the study was done? (Justification)

2. Add a clear hypothesis of the study.

Reviewers' comments:

Reviewer's Responses to Questions

**Comments to the Author**

1. If the authors have adequately addressed your comments raised in a previous round of review and you feel that this manuscript is now acceptable for publication, you may indicate that here to bypass the “Comments to the Author” section, enter your conflict of interest statement in the “Confidential to Editor” section, and submit your "Accept" recommendation.

Reviewer #2: All comments have been addressed

2. Is the manuscript technically sound, and do the data support the conclusions?

Reviewer #2: Yes

3. Has the statistical analysis been performed appropriately and rigorously? 

Reviewer #2: Yes

4. Have the authors made all data underlying the findings in their manuscript fully available?

Reviewer #2: Yes

5. Is the manuscript presented in an intelligible fashion and written in standard English?

Reviewer #2: Yes

6. Review Comments to the Author

Reviewer #2: Thank you for proving a revised version of the manuscript. All my comments were properly addressed. I have no new comments.

7. PLOS authors have the option to publish the peer review history of their article (what does this mean?). If published, this will include your full peer review and any attached files.

Reviewer #2: **Yes: **Arthur de Sá Ferreira

---

## [Author Response · Author response to Decision Letter 1]

16 Aug 2022

Response to comments from the editor and reviewers - Manuscript PONE-D-21-38018R1

We appreciate the time invested by the Editor and reviewers for their thoughtful comments and for the opportunity given to improve our manuscript. We provide below the detailed answers to each comment.

Academic Editor:

1. The rationale of the study should be explained in detail. You should focus on three elements of introduction:

a. What is known about the topic? (Background)

b. What is not known? (The research problem)

c. Why the study was done? (Justification)

We have completed and reorganized the introduction by cutting it as suggested:

- Background (p2, lines 29-52);

- The research problem (p2-3, lines 52-57);

- Justification (p3, lines 58-61).

The introduction now quotes :

Background :

 « Musculoskeletal disorders (MSKDs) are a major public health concern worldwide and represent globally the second most important group of disorders in terms of years lived with disability (1). The most common MSKDs encountered in primary care are spinal pain including neck, thoracic spine or low back disorders (2,3). For the majority of patients consulting for spinal pain in primary care, there is strong evidence supporting the benefit of early physiotherapy in the care pathway for these patients (4,5). However, this early access is not always carried out as systematically and smoothly as intended. The consequences are delayed treatment and potential clinical complications such as chronic pain (6–8).

Depending on the country, access to primary care is organized or constrained in different ways. In some health care systems, general practitioners (GPs) are first-contact providers for patients seeking care for spinal pain complaints. GPs have a key role in the patient’s care pathway by providing an initial diagnosis and treatment and referring to other appropriate health care professionals such as physiotherapists (PTs) (9). The purpose of this referral is to validate the indication for physiotherapy care and to identify any contraindications or precautions to rehabilitation for a specific patient (9). However, several studies have concluded that diagnoses provided by GPs for this population may often be erroneous or not as accurate as those provided by other MSKD specialists such as sports physicians, orthopedic surgeons or even PTs (10–12). In addition, GPs’ practice patterns in terms of treatment recommendations have been reported to divert significantly from established evidence-based clinical practice guidelines (CPGs); poor patient education as well as poor promotion of active treatments such as physical activity is often reported (6,7,13–18). An Australian study showed that the percentage of patients seeing a GP for MSKDs who were referred to physiotherapy was low out of the total number of patients seen by the physician with the same condition (19). An observational study in Denmark reported that, for patients with various MSKDs, GPs' primary diagnoses were generally poorly defined with the use of vague terms such as myopain or back-related diagnoses (20). 

France belongs to those countries where the GP is the gatekeeper to the health care system and might secondarily refer a patient with MSKDs to a PT. MSKDs account for approximately 17.0% of French GPs consultations (13,21) and their incidence is expected to increase as the French population is ageing rapidly (22). »

Research problem :

“French GPs’ physiotherapy referral practices for patients with spinal pain complaints in primary care have not been described and reported so far. Thus, the extent to which French GPs and PTs practices as regards to physiotherapy interventions prescribed, are supported by evidence-based recommendations of CPGs is not known. Moreover, to our knowledge, no study has been conducted to compare patient’s MSK diagnostic and physiotherapy interventions concordance between GPs and PTs with respect to CPGs.”

Justification : 

“In a system where the GP is the patient's first access, even if it is expected that his/her knowledge of evidence-based practice is up to date, it is likely that this knowledge cannot be exhaustive on such a specific subject, due to the polyvalence of this specialty. Similarly, it is expected that the skills acquired by the PT profession are at a high level of expertise and therefore in line with CPGs.”

2. Add a clear hypothesis of the study.

An hypothesis has been added (p3, lines 61-64) :

“This study assumes that the characteristics of referral to physiotherapy described by French GPs for patients with spinal pain are incompletely reported, as in other countries, and that physiotherapy interventions recommended by PTs could be significantly more consistent with CPGs than those prescribed by GPs.”

---

## [Editor Report · Decision Letter 2]

22 Aug 2022

Spinal pain patients seeking care in primary care and referred to physiotherapy: A cross-sectional study on patients characteristics, referral information and physiotherapy care offered by general practitioners and physiotherapists in France

PONE-D-21-38018R2

Dear Dr. DEMONT,

We’re pleased to inform you that your manuscript has been judged scientifically suitable for publication and will be formally accepted for publication once it meets all outstanding technical requirements.

Kind regards,

Walid Kamal Abdelbasset, Ph.D.

Academic Editor

PLOS ONE
---

## [Editor Report · Acceptance letter]

26 Aug 2022

PONE-D-21-38018R2 

Spinal pain patients seeking care in primary care and referred to physiotherapy: A cross-sectional study on patients characteristics, referral information and physiotherapy care offered by general practitioners and physiotherapists in France 

Dear Dr. DEMONT:

I'm pleased to inform you that your manuscript has been deemed suitable for publication in PLOS ONE. Congratulations! Your manuscript is now with our production department. 

Kind regards, 

on behalf of

Dr. Walid Kamal Abdelbasset 

Academic Editor

PLOS ONE